# Positional Analysis of Assisting Muscles for Handling-Assisted Exoskeletons

**DOI:** 10.3390/s24144673

**Published:** 2024-07-18

**Authors:** Zheng Wang, Xiaorong Guan, Long He, Meng Zhu, Yu Bai

**Affiliations:** 1School of Mechanical Engineering, Nanjing University of Science and Technology, Nanjing 210094, China; wangzheng19@njust.edu.cn (Z.W.); helong@zy-cs.com.cn (L.H.); zhumeng0120@njust.edu.cn (M.Z.); byu101010014@njust.edu.cn (Y.B.); 2Zhiyuan Research Institute, Hangzhou 310000, China

**Keywords:** Anybody Modeling System, assisted exoskeletons, surface electromyography, muscle activation

## Abstract

In order to better design handling-assisted exoskeletons, it is necessary to analyze the biomechanics of human hand movements. In this study, Anybody Modeling System (AMS) simulation was used to analyze the movement state of muscles during human handling. Combined with surface electromyography (sEMG) experiments, specific analysis and verification were carried out to obtain the position of muscles that the human body needs to assist during handling. In this study, the simulation and experiment were carried out for the manual handling process. A treatment group and an experimental group were set up. This study found that the vastus medialis muscle, vastus lateralis muscle, latissimus dorsi muscle, trapezius muscle, deltoid muscle and triceps brachii muscle require more energy in the process of handling, and it is reasonable and effective to combine sEMG signals with the simulation of the musculoskeletal model to analyze the muscle condition of human movement.

## 1. Introduction

The manual handling process usually includes a series of movements, such as handling, bending, grasping, returning to the upright position, turning and handling the arms. In the absence of auxiliary equipment protection, manual handling is prone to muscle fatigue due to excessive work intensity or unstandardized handling movements, which can lead to acute musculoskeletal injuries and local pathological changes in severe cases [1,2,3]. The purpose of handling-assisted exoskeletons is to provide assistance to the human handling process while reducing muscle fatigue and the risk of musculoskeletal injuries. Therefore, before carrying out the design of an exoskeleton, it is necessary to analyze and study the movement characteristics and injury mechanism of the musculoskeletal joints involved in the handling process to ensure the rationality of the exoskeleton design.

Most of the existing biological analyses for handling exoskeleton design remain at the theoretical stage. Most of the existing exoskeletons analyze the joint forces during human movement through mechanics, without analyzing the specific muscle movements. In 2022, Tröster performed a biomechanically model of human motion for better exoskeletons design [4]. Occupational exoskeletons are wearable devices designed to assist and support workers in physically demanding tasks. They aim to reduce the strain and stress on the human body by providing mechanical support and improving the human body. Biomechanical models play a significant role in understanding the effects of occupational exoskeletons on the human body. These models incorporate detailed anthropometric measurements, such as body dimensions, joint ranges of motion, and muscle strength, to simulate and analyze the interaction between the exoskeletons and the wearer. The different kinds of existing exoskeletons analyzed include HAL−5 from the University of Tsukuba [5], HEXAR from Hanyang University [6], PERCRO BodyExtender from Scuola Superiore S. Anna [7], AXO–SUIT designed by Aalborg University [8], Denmark, Berkeley Lower Limb exoskeletons (BLEEX) designed with support from the Defense Advanced Research Projects Agency (Berkeley, capital of New York City), USA [9,10], etc. Comparison of exoskeletons’ booster modes and strapping positions reveals that the booster modes and strapping positions of exoskeletons with different roles are different, but there has been no experimental or simulation analysis of this, so it is necessary to analyze the booster modes and strapping positions through experiments and simulations. AMS and sEMG signals can be used to analyze the condition of human muscles during movement, which can help the exoskeletons to choose a better booster mode and tethering position.

Many biomechanical models have been developed to better study human motion. With the continuous development of computer technology and dynamics theory, by the 1990s, the theory and methods of multi-rigid body dynamics and rigid–flexible coupled dynamics began to be applied to the modeling study of human bone–muscle system dynamics. Since entering the 21st century, more and more new theories and methods have appeared in the field of human bone and muscle system dynamics modeling and simulation research. Musculoskeletal modeling, consisting of software such as the Anybody Modeling System (AMS), uses mathematical modeling techniques to simulate the diverse muscles and bones in the human musculoskeletal structure. The ratio between the muscle contribution and its corresponding strength, that is the muscle activation level, has been used to compare the acceptability of alternative workplace layouts [11,12,13,14,15].

Zhou and Wiggerman estimated the loads on the lumbar spine joints using AMS and verified them experimentally by combining simulations and experiments to jointly verify the loads on the lumbar spine joints in relation to the patient’s body weight [16]. Smith et al. proposed an exoskeleton human musculoskeletal model to analyze upper limb motion, which was driven exclusively with data from 3D motion capture to analyze shoulder and elbow forces and moments [17]. Yong–Ku Kong et al. evaluated wearing a lower limb exoskeleton based on EMG experiments and AMS simulation to analyze the loading of the lower limb muscles during a bolus task. They indicated that AMS simulation assessment can be used instead of EMG analysis. In particular, AMS is an inverse dynamic biomechanical analysis program that is often used in the study of the human body to simulate the use of exoskeletons [18]. Thus, AMS can be used in conjunction with surface EMG to analyze human movement.

At this stage, the biomechanical analysis of the human body under specific movements is more often analyzed by using AMS software (Anybody 6.0) alone or in combination with joint motion data, but this method is not supported by the subsequent experimental data and is not analyzed together with the experimental results. In this study, AMS simulation results and sEMG data are combined to analyze the biomechanics of the human body under specific movements, which makes the analysis results more convincing.

Muscles produce changes in sEMG signaling during human movement, and different actions correspond to changes in sEMG signaling in different muscle combinations, which establishes a relationship between sEMG signaling and human movement. Recently, the concept of muscle synergy has been investigated in order to estimate muscle activation through static optimization (SO) or EMG-driven modeling of human musculoskeletal bones [19,20,21,22]. Muscle synergy consists of a time-varying synergistic excitation and a corresponding time-varying synergistic vector, which contains weights that define the contribution of muscle synergy to all muscle activations [23,24,25]. Muscle synergism can be used to represent a large number of measured or simulation-modeled muscle activations with a small number of muscle synergisms, as well as to study the effects of changes in muscle weight on other muscles [26]. The addition of muscle synergies to muscle activations estimated by Michaud et al. and Shourijeh and Fregly for the SO model requires that muscle activations be solved for all time frames simultaneously [21,22]. In both studies, the muscle–tendon model parameter values were not calibrated simultaneously, EMG experiments were not used as a basis for muscle activation, and the accuracy of predicting muscle activation was no better than that of the original SO model when compared to the results of EMG experiments. In another recent study, Ao et al. predicted synergistic vector weights for unmeasured muscle activation by simplifying the EMG-driven musculoskeletal modeling process using synergies calculated from muscle activation measured by EMG experiments [20]. To assess the feasibility of this approach (referred to as “synergistic extrapolation” or SynX), the authors used an EMG-driven musculoskeletal model whose activations and muscle–tendon model parameters had been modeled and calibrated using a complete set of EMG experimental measurements of subjects’ movement data. The accuracy of this model was demonstrated by subsequent experimental results [27]. These studies show that the analysis of human motion should combine musculoskeletal model simulation and EMG experiments and that such a treatment can improve the accuracy of the analysis of human motion.

The main objective of this study is to identity the muscles that need to be supported in handling-assisted exoskeletons using AMS simulation software (Anybody 6.0) and sEMG experiments. Based on the Anybody Modeling System, the muscle activation of some muscles of the upper and lower limbs in weight-bearing handling was simulated and compared with that in non-weight-bearing handling, and the muscles whose muscle activation increased significantly after increasing the weight–bearing were identified. The sEMG experiments were carried out for handling with different loads, and the muscles with a higher muscle contribution during handling were obtained by processing and analyzing the sEMG signals. Through the joint analysis of the simulation results and experimental results, the muscles that really need assistance in the handling process are found to provide theoretical support for the subsequent design of handling-assisted exoskeletons.

## 2. Materials and Methods

### 2.1. Muscle Selection

The human body relies on the trunk and upper and lower limbs to perform handling movements, in which the hip joint is an important joint that plays a role in connecting, and the health of the hip joint will directly affect the human body in performing various movements. Hip flexor muscles include iliopsoas muscle, quadriceps muscle, sartorius muscle, pubococcygeus muscle, longissimus muscle, shortissimus muscle, etc. Among them, the quadriceps muscle and the anterior psoas muscles have the greatest strength. As well as flexing the hip joint, the quadriceps muscle is also responsible for straightening the knee joint. In addition, the broad fascia tensor muscle for hip flexion also has a certain role, flexion of the hip joint muscle being its main role in addition to flexion of the hip joint, but more importantly, to maintain the balance of force around the hip joint, human posture and walking gait are very important.

The squat requires foot stabilization, ankle dorsal flexion, knee flexion, hip flexion, slight posterior pelvic tilt, spinal neutrality and maintaining body balance. These are the basic movements of the joints during a squat. However, it is not only the joints that are involved in a squat but also the nerves, muscles, ligaments, respiration and all the structures. Damage to nerves or muscles, or prolonged holding of a relatively fixed position, will affect the sequence of movements during the completion of the movement, which in turn will affect the position of the joints and the respiratory position. Therefore, it is very necessary to carry out exoskeleton assistance for handling movement, and it is also necessary to carry out human movement research for handling movement.

The muscles required for the human body to perform the handling action are relatively many, but due to the specificity of the exoskeletons’ assisting position, the assisting muscles belong relatively to the main muscle groups, and they are in a better position to not affect the human body’s movement. Therefore, in this study, the muscle selection rules for the analysis of human movement come from the main active muscle groups that are convenient for the exoskeletons to improve the power or load transfer, including the obliques, biceps, and triceps in the upper limb arm muscles; the rectus femoris, medial femoris, lateral femoris, vastus tensor fasciae latae, and biceps femoris in the thigh muscles of the lower limbs; and, finally, the latissimus dorsi muscle and the external oblique muscle in the lumbar and abdominal muscles. The muscles of the lumbar and abdominal muscles are also included.

### 2.2. Simulation of Musculoskeletal Model

The AMS uses inverse dynamics analysis to calculate the muscle forces required to maintain the manikin in a specified pose under external loads. The strength of the dummy is determined based on the maximum contribution of at least one muscle. The strength of the manikin is, therefore, dependent on the individual strength of each muscle in the body and the muscle recruitment criteria that determine the level of activation of each muscle in a given task.

To accurately represent the behavior and strength of real muscles, the human model takes into account the physiological cross-sectional area of each muscle. These physiological cross-sectional area values are obtained from various cadaver studies. In addition, different levels of detail are available in the software to simulate the working behavior of the muscle model, specifically taking into account the effects of working conditions on the ideal strength of the muscles. The 3-element model, based on the modified Hill muscle model introduced by Zajac, is the most detailed muscle model in the software.

This simulation mimics the human body’s handling action of placing items in the hand on the floor for analysis. To build the required model, users have the option of starting from scratch or modifying an existing model from the model library. In this case, a full body model was chosen, incorporating a center-of-mass driver to maintain balance and predicting ground reaction forces for contact with the ground. The existing model was then modified to include the handling movement, with the establishment of a treatment group and an experimental group.

The tests in this study were conducted from human standing and tested the entire lifting process, starting with the experimenter standing ready to go, then performing a squat, and then standing up for a complete cycle of handling action. The experimental group included, in addition to handling, the application of a 50 N force in each hand to simulate the weight being lifted during the movement. The analysis was performed using inverse dynamics, with the number of steps set to 1000. The model was executed in the chosen language to conduct the required analysis. In this study, the three motion cycles of the handling action are taken as a set of simulation motions, the set of simulation motions is divided into 1000 steps, and the data extraction is performed twice in each step, which will eventually result in 2000 consecutive data curves.

By incorporating these modifications and running the inverse dynamics analysis, the model allows for the evaluation of the biomechanics and forces involved in the handling motion, considering the impact of additional weight in the experimental group. This provides insight into the muscle activation, joint forces, and overall movement patterns during the handling motion, aiding in the analysis and design of handling-assisted exoskeletons.

Based on the analysis of the human kinematics of the human body performing the handling action, the muscles that are mainly used in the handling process are counted. Then, among these muscles, we find that the muscles that can be assisted by covering the exoskeletons and the sEMG data are relatively good to measure, and we find the 11 muscles in Table 1 in these two conditions. The exact location of these muscles is shown in Figure 1.

### 2.3. Experiment of Surface Electromyography

The subjects were 10 young men aged 22–26 years old. The subjects were healthy, 165–185 cm in height and 60–80 kg in weight, and were required to be right-handed with no history of muscle strain and to have a habit of daily exercise. The subjects were required to have no history of muscle strain and no strenuous exercise in the 72 h before the test to minimize the influence of muscle fatigue on the results of this test.

This experiment was approved and consented to by the Ethics Committee of the First Affiliated Hospital of Nanjing Medical University. And informed consent was obtained from all subjects before the experiment. All methods comply with the relevant provisions of the Civil Code of the People’s Republic of China on human experimentation.

The experiment was divided into experimental group and treatment group. The experimental group required the subjects to squat and reciprocate with 10 kg goods in their hands, and the treatment group required them to reciprocate without heavy goods in their hands; the experimental process was the same in both groups except for this variable. During the experiment, the subjects were required to remain upright for one second while squatting and standing, ten times in each group. The same subjects were used in the experimental group and the treatment group, but the subjects were required to perform the experiment across two days in order to minimize the error caused by muscle fatigue.

The sEMG experiment applies a set of wireless sEMG electrode patches, and the experimental equipment is shown in Figure 2.

sEMG experiments for muscle selection, taking into account the upper limb muscle groups in the case of hand weight-bearing certainly being the case for larger surface EMG signals, do not need the experimental process on which the muscle activation degree is calculated again in the case. Thereby, the experimental measurement muscle selection is shown in Table 2.

Because the ambient temperature has a certain effect on the size of the sEMG signal, the ambient temperature was set to 24 °C. The subjects were warmed up for 2 min, and then the skin near the placement point of the electrode and the placement point were shaved; after dehairing, the skin was wiped with a wet towel, the skin was repeatedly wiped with a wet towel with 75% medical alcohol, and then after the skin was dry, the patch electrodes were attached in the middle of the muscle abdomen of the target muscle, and the patch electrodes were placed in position, as shown in Figure 3.

## 3. Results

### 3.1. Inverse Dynamics Analysis

The muscle recruitment criteria in the Anybody Modeling System aim to emulate the functioning of the central nervous system (CNS) in selecting the appropriate muscles for a given task. As the human body possesses more muscles than necessary to perform a specific task, accurately replicating the CNS-selected criteria, especially for the utilization of antagonistic muscles, can be challenging. For instance, co-contraction of antagonistic muscles is required to stabilize joint movements and is influenced by the desired motor precision [28]. Given that each muscle is divided into multiple parts within the Standing Model, muscle activation (MVC) needs to be averaged across all parts of a muscle for analysis and simulation purposes.
(1)$$MVC=\left(\sum_{i=1}^{n}MVC_{i}\right)/n$$
where *n* denotes the number of corresponding muscles in the model, and *MVC_i_* denotes the muscle activation degree of each part. The final muscle activation data of the 11 muscles on the right half of the human model are derived. The final muscle activation data for the 11 muscles in the treatment group are shown in Figure 4.

### 3.2. sEMG Signal Pre–Processing

The sEMG signal is an unstable signal, and, the process of acquisition, external environmental interference, signal acquisition equipment and the human body itself will generate noise. The sources of these noises and the concentration frequency distribution are shown in Table 3 [29].

In view of the noise interference mentioned above, this study performs noise reduction processing on the signal. Specifically, a 4th-order Butterworth filter is utilized for filtering purposes. The filter is configured as a 4th-order bandpass filter, with frequency parameters set between 20 Hz and 500 Hz. The Butterworth filter is characterized by a maximally flat frequency response curve within the pass band, exhibiting no rise or fall. In the stop band, the response gradually decreases towards zero. In the Porter diagram, which represents the logarithm of the angular frequency amplitude, the amplitude diminishes as the corner frequency increases, eventually approaching negative infinity. The amplitude diagonal frequency of the Butterworth filter follows a monotonically decreasing pattern and maintains the same shape for different filter orders. However, higher-order Butterworth filters exhibit faster amplitude decay in the stop band compared to lower-order filters. Other types of filters display different amplitude diagonal frequency curve shapes for higher orders compared to lower orders [30,31,32].

A comparison of raw and pre-processed data of surface EMG signals is shown in Figure 5. Surface EMG signal data from 12 pre-processed muscles of the same tester are shown in Figure 6.

### 3.3. sEMG Signal Feature Extraction

In this study, the raw sEMG signals acquired from the collector are multichannel continuous non-stationary time series. Each time point in the signal represents the current amplitude of the sEMG signal. While the sEMG signal contains valuable information for classification and identification purposes, the differences between different actions cannot be directly used by the classifier for classification. Therefore, it is necessary to process the original signal and extract the most representative components, a process known as feature extraction.

Time–domain characterization involves considering the sEMG signal as a function of time over a short period and calculating performance indicators based on the signal’s amplitude. This method has been widely applied in myoelectric signal identification research due to its inherent advantages, such as convenient calculation without complex transformations, ease of implementation, and good generality. Although time–domain features can easily extract representative features, researchers have observed that the non-stationarity of time–domain methods is significant. Even slight changes in muscle contraction force during the signal acquisition process can lead to fluctuations in time–domain features. To address this instability, researchers have explored frequency–domain feature analysis methods, which involve analyzing the spectral distribution characteristics of the signal through frequency spectrum and power spectrum analysis.

By employing both time–domain and frequency–domain feature extraction methods, researchers aim to capture essential information from the sEMG signals and improve the accuracy and reliability of the subsequent classification and identification tasks [33,34,35,36].

The waveform length (WL) is a measure of the complexity of the signal that can be obtained by calculating the difference in amplitude change between each adjacent sample point of the sEMG signal over a certain period of time and summing up the accumulation of all signal differences in that segment. The expression is as follows:(2)$$WL=\sum_{i=1}^{N-1}\left|x_{i+1}-x_{i}\right|$$
where *N* denotes the length of the sampling window, and *x_i_* and *x_i+_*_1_ denote the signal values of two neighboring sampling points, respectively.

The amplitude of the acquired sEMG signal varies when different limb movements are performed, and the absolute mean value can describe the average intensity of the EMG signal, which is calculated as shown below.
(3)$$MAV=\frac{1}{N}\sum_{i=1}^{N}\left|x_{i}\right|$$
where *x_i_* is the sEMG signal and *N* is the length of the sample data.

The root mean square is able to describe the effective value of the signal, and, to a certain extent, it can reflect the size of the contribution of each muscle to the limb movement; its calculation formula is shown as follows.
(4)$$RMS=\sqrt{\frac{1}{N-1}\sum_{i=1}^{N}{x_{i}}^{2}}$$
where *x_i_* is the sEMG signal and *N* is the length of the sample data.

The average power frequency is the frequency that falls at the center of the signal power spectrum curve, with the following equation.
(5)$$MPF=\frac{\int_{0}^{\infty }f\cdot PSD(f)df}{\int_{0}^{\infty }PSD(f)df}$$
where *f* is the frequency of the sEMG signal, and *PSD*(*f*) is expressed as the power spectral density function of the surface EMG signal at frequency f. The expressions are:(6)$$PSD(f)=\frac{1}{T}{\left|x(k)\right|}^{2}$$

The curves of the four eigenvalues of the same muscle are shown in Figure 7.

### 3.4. Results of the Simulation

In the AMS simulation results for the handling movement, the muscle activation values represent the average activation of all parts of each muscle in the software. This average activation reflects the overall muscle activation during the handling movement to some extent. By comparing the muscle activation values of the 11 identified muscles, it is possible to determine which muscles exhibit higher levels of activation during the handling movement.

This comparison helps identify the muscles that are more actively involved in the movement and may require assistance or support from an exoskeleton. By focusing on these muscles with higher activation levels, researchers can gain insights into the specific muscles that are crucial for performing the handling movement and potentially benefit from the assistance provided by an exoskeleton in those areas. This analysis contributes to the understanding of muscle activation patterns and assists in the subsequent design and development of exoskeletons tailored to augment or support specific muscle groups during handling tasks.

Figure 8 illustrates a comparison of muscle activation during the handling action for two different hand-applied forces by analyzing the changes in the standard deviation of muscle activation for each individual muscle. Additionally, the muscle activation data for the 11 muscle groups were integrated, and the results were compared between the two groups. The results of the comparison of MVC integral values and standard deviation between the treatment group and the experimental group are shown in Figure 9.

By analyzing the integral values of muscle activation for the control and experimental groups, several notable findings can be observed. Firstly, muscle No. 9 in the experimental group exhibited the highest integral value of 349.1, representing a substantial increase of 300% compared to the treatment group. This suggests that the muscle activation in this particular muscle was significantly enhanced in the experimental group. Additionally, muscles No. 8 and No. 10 also displayed considerable increases in integral values. Muscle No. 8 exhibited a 168% increase in activation, while muscle No. 10 demonstrated a substantial 360% increase. These findings indicate that the experimental group experienced pronounced enhancements in the activation of these muscles compared to the treatment group. Moreover, muscles No. 2, No. 3, and No. 6 also exhibited significant increases in integral values. Although the specific percentage increases are not provided, these findings suggest that these muscles were also positively affected by the experimental conditions, showing higher levels of activation compared to the treatment group. Overall, a comparison of integral values highlights specific muscles that demonstrated notable increases in activation in the experimental group. These findings help to explain the impact of different hand-applied forces on muscle activation during the handling action.

Upon comparing the standard deviation results of the control and experimental groups, several noteworthy observations were made. Notably, muscle No. 10 in the experimental group exhibited the highest standard deviation value of 0.077, which represented a substantial increase of over 600% compared to the treatment group. This indicates that the variability in muscle activation was significantly higher in the experimental group for this particular muscle. Furthermore, muscles No. 6, No. 8, and No. 9 also displayed significant increases in their standard deviation values. Although the specific percentage increases are not provided, these findings suggest that the experimental group experienced notable enhancements in the variability of muscle activation compared to the treatment group for these muscles. Additionally, muscles No. 2, No. 3, and No. 4 also demonstrated significant increases in their standard deviation values. This suggests that the experimental group exhibited higher levels of variability in muscle activation compared to the treatment group for these specific muscles. In summary, a comparison of standard deviation values reveals specific muscles that showed significant increases in the variability of activation in the experimental group. These findings provide insights into the impact of different hand-applied forces on the variability of muscle activation during the handling action.

### 3.5. Results of the Experiment

The frequency-domain eigenvalues of sEMG signals provide valuable insights into the level of muscle exertion, with larger eigenvalues indicating greater muscle activity. To assess the degree of muscle exertion, the sEMG data underwent pre-processing, and four different eigenvalues from the same participant were integrated. The integration results were then compared to the maximum values for both the control and experimental groups. Furthermore, the integration results of the four eigenvalues from everyone were averaged across all participants. Subsequently, the averaged results were integrated once again, and the integration results were compared to the maximum values for both the control and experimental groups. By performing these analyses, a comprehensive understanding of the degree of muscle exertion in both groups can be obtained. Comparing the integration results to the maximum values provides insights into the overall level of muscle activation, allowing for a comparison between the control and experimental groups.

Figure 10 compares the maximum values of the four eigenvalues from the same participant, and the results show that there is no clear pattern or significant difference between the eigenvalues. However, in Figure 10, when comparing these values to those of other muscles, it can be observed that muscles No. 7, No. 8, No. 9, and No. 10 tend to have larger eigenvalues. On the other hand, a comparison of the integration results reveals more noticeable differences. In the experimental group, all four eigenvalues show significantly higher values compared to the treatment group. Specifically, muscles No. 7, No. 8, No. 9, and No. 10 exhibit a significant increase across all four eigenvalues. Additionally, muscles No. 1 and No. 2 show a significant increase in the three frequency–domain eigenvalues of MAV, RMS, and WL. These findings indicate that the experimental group experienced a higher degree of muscle exertion, as evidenced by the increased values in the integrated eigenvalues across multiple muscles.

In the comparison between the control and experimental groups, after averaging the maximum values of the four eigenvalues across all participants, clear patterns emerged in the comparison of the integrated eigenvalues. The comparison results are shown in Figure 11. In the frequency–domain eigenvalues MAV, RMS, and WL, muscles No. 1, No. 2, No. 9, and No. 10 exhibit larger values, and there is a significant increase in these muscles in the experimental group compared to the treatment group. Specifically, muscles No. 1 and No. 2 show a growth of over 100% in these eigenvalues. Regarding the frequency–domain eigenvalue MPF, all muscles in the experimental group demonstrate a significant increase compared to the treatment group. In the comparison of the maximum values, the time–domain eigenvalue MPF shows significant fluctuations, without any clear trend. However, the frequency–domain eigenvalues MAV, RMS, and WL display more pronounced patterns. Muscles No. 1, No. 2, No. 9, and No. 10 have larger maximum values, and the experimental group shows a significant increase of over 90% compared to the treatment group.

### 3.6. Results

In the simulation results, integrating the integral value and standard deviation of muscle activation (MVC), it can be seen that the muscle activation of No. 2, No. 3, No. 6, No. 8, No. 9 and No. 10, i.e., medial femoral, lateral femoral, latissimus dorsi, trapezius, deltoid, and triceps, is higher during weight-bearing handling, and these muscles are more in need of a booster during weight-bearing handling. In the experimental results, combining these results, it can be obtained that No. 1, No. 2, No. 9 and No. 10, i.e., bilateral latissimus dorsi and bilateral medial femoral muscles, have higher muscle effectiveness and require more assistance during weight-bearing handling.

Combining the simulation and experimental results, the following conclusions can be drawn: In the process of handling without load, the muscle activation of the three anterior thigh muscle groups, namely the medial femoral muscles, lateral femoral muscles, and rectus femoris, was higher. After increasing the load, in addition to the shoulder muscles of the upper limb, the activation of the latissimus dorsi muscles increased the most, which should pay more attention to the comfort of the back as well as the effect of the assisting force in the subsequent design of the exoskeletons. In the handling experiments after adding load, it can be seen that the lower limb muscle activation was higher in the process of weighted handling.

In the handling experiment after increasing the load, it can be seen that the muscle activation of the lower limb muscles of the medial femoral muscle and lateral femoral muscle increased more, and the frequency–domain eigenvalue of the sEMG also increased more, but we should not neglect the rectus femoris muscle, which has a larger muscle activation, and other values when there is no load. From the experiments, it can be seen that during handling, the rectus femoris muscle has a larger relative value, so in the design of the exoskeletons, in addition to the consideration of the medial femoral muscle and lateral femoral muscle, it should also take into account the assistance of the rectus femoris muscle.

## 4. Discussion

This study found that the medial femoral, lateral femoral, latissimus dorsi, rhomboids, deltoids, and triceps brachii muscles needed more assistance during the handling process, while the lumbar and abdominal muscles did not need to be considered as a priority, in which the latissimus dorsi muscle should be considered regarding the exoskeletons’ effectiveness in assisting it, and it can be appropriate to increase the assistance to the rectus femoris muscle. This study combines the simulation of the musculoskeletal model and the experiment of sEMG to show that the muscle condition of the human movement process is reasonable and accurate, and this method can be applied to similar studies.

## 5. Conclusions

Due to the limited number of channels of the sEMG equipment, the muscles selected in the sEMG experiments in this study did not completely cover the muscles selected in the Anybody software (Anybody 6.0), but it is more necessary to pay attention to the effect of the lower limb muscle group booster effect in the design of the exoskeletons, so the present experiments can prove the accuracy of the Anybody software results to a certain extent. However, the subsequent ability to complement the Anybody simulation results of the muscles may provide better results.

This study simplifies the handling process to weighted handling, which can reflect the real results in the vertical direction, but it does not consider actions such as turning and handling arms, and it will be better to add the consideration of these actions in a subsequent study. In addition, in the weight design of weighted handling, there were no more groups; only no weight and 10 kg weight were used in this study, and there was no grading of the loads, which might produce different results under different loads, and this part of the study can be added in future research.

This study is a biomechanical analysis of the exoskeletons prior to their design, but whether or not the actual design of the subsequent exoskeletons results in a complete reflection of the desired state needs to be established in subsequent research. By testing the actual booster effect of the exoskeletons, it can reflect whether the booster position of the human body is correct or not, and this aspect of the research needs to be continued in subsequent studies.

## Figures and Tables

**Figure 1 sensors-24-04673-f001:**
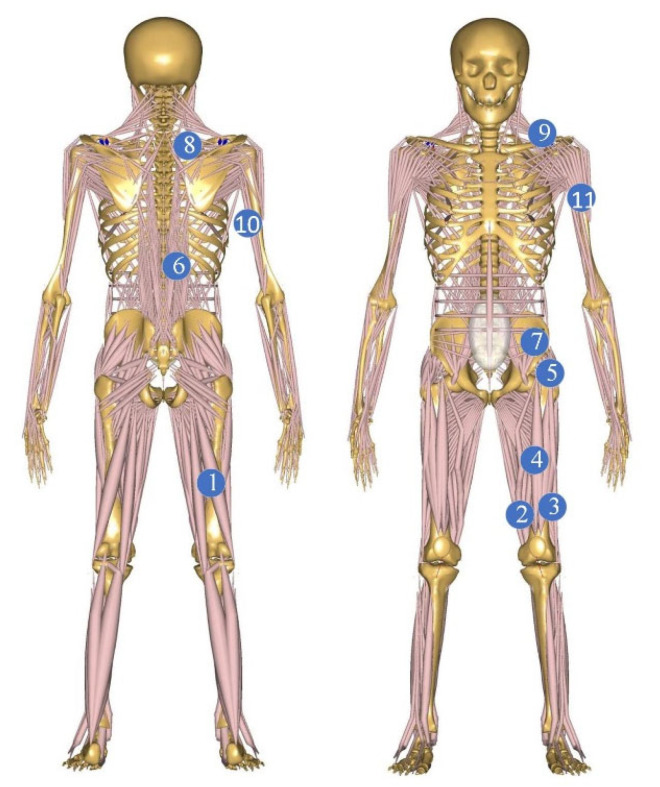
Model handling process and selection of muscle position.

**Figure 2 sensors-24-04673-f002:**
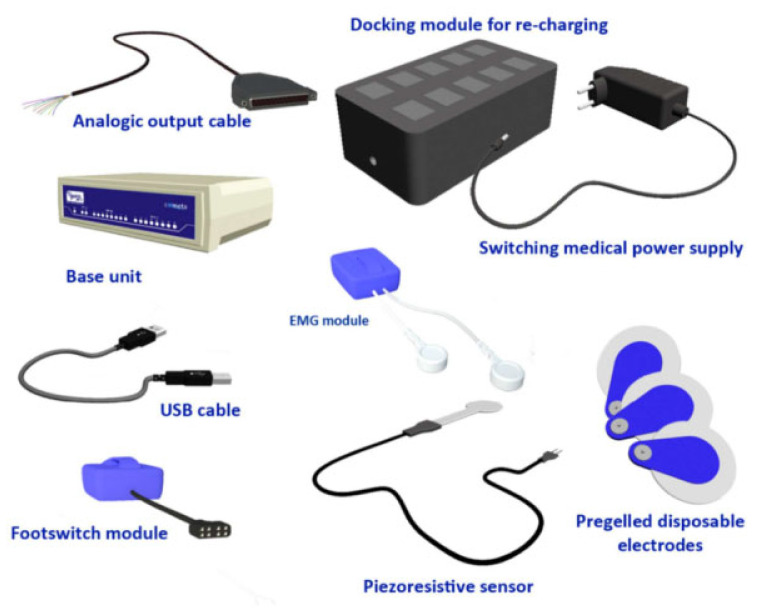
Typical Wave Plus components.

**Figure 3 sensors-24-04673-f003:**
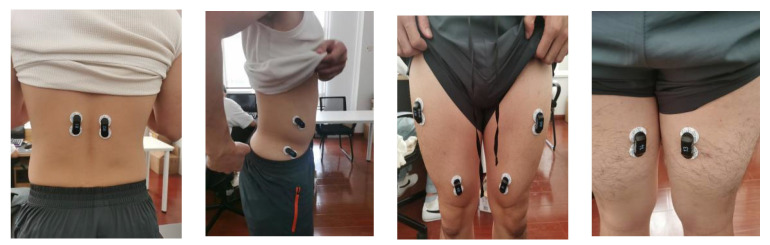
Placement of chip electrodes.

**Figure 4 sensors-24-04673-f004:**
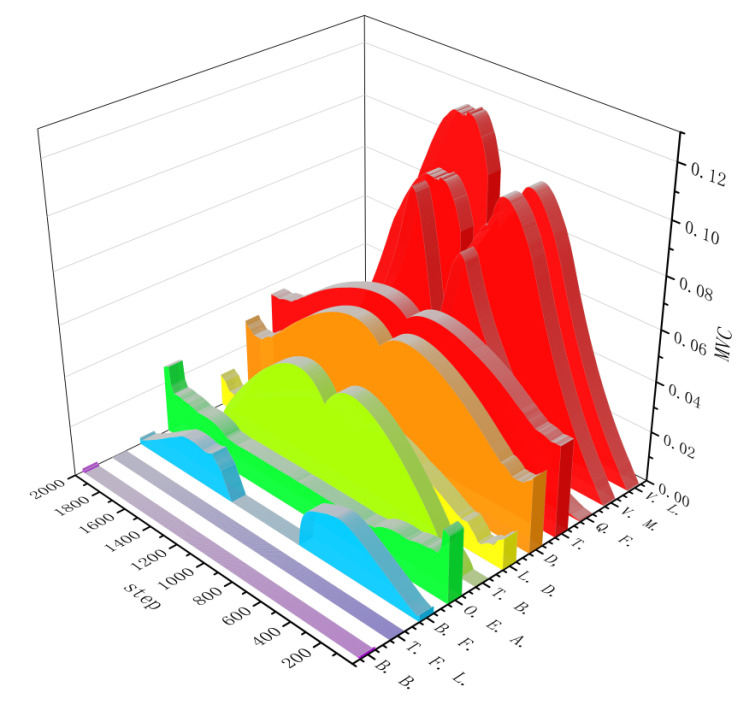
Muscle activation of the 11 muscles in the treatment group.

**Figure 5 sensors-24-04673-f005:**
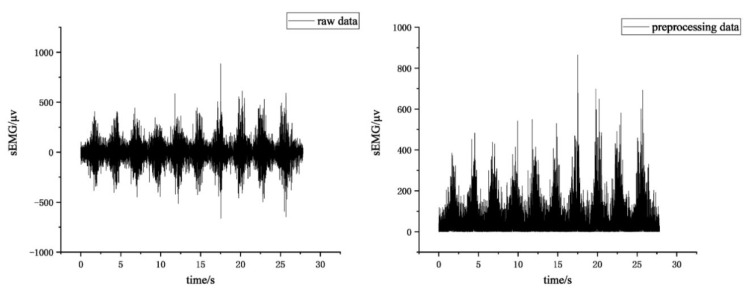
Raw and pre-processed data of sEMG.

**Figure 6 sensors-24-04673-f006:**
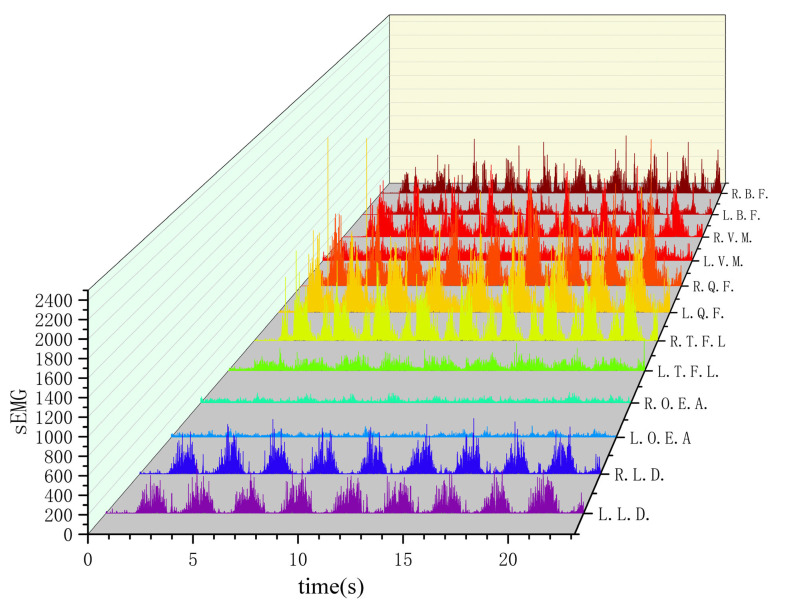
sEMG signals from 12 muscles of the same tester after pre-processing.

**Figure 7 sensors-24-04673-f007:**
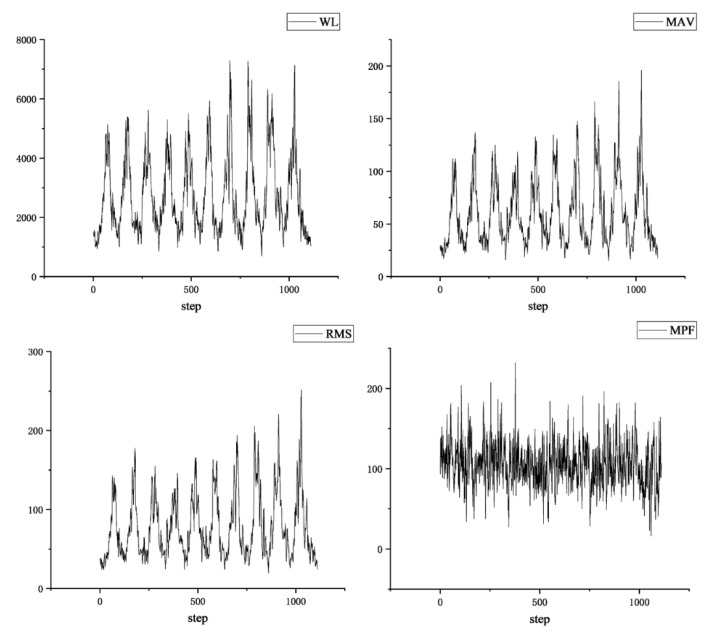
Four eigenvalues of the same muscle.

**Figure 8 sensors-24-04673-f008:**
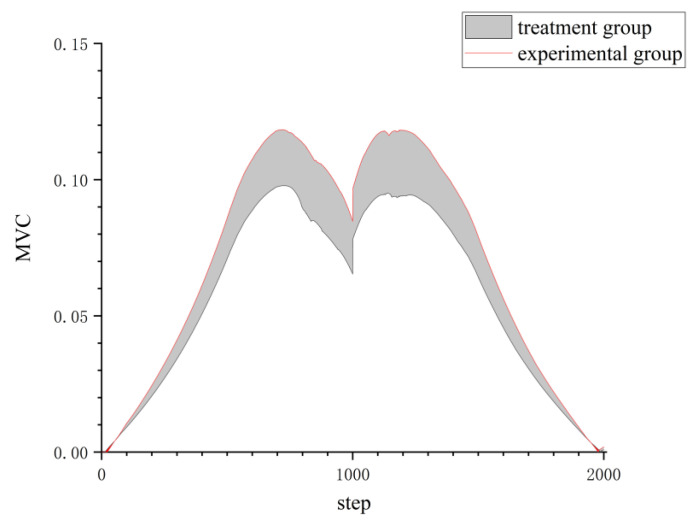
Muscle activation of medial femoral muscle treatment group compared with experimental group.

**Figure 9 sensors-24-04673-f009:**
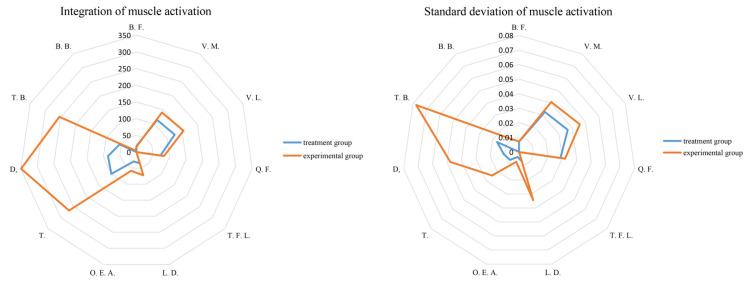
Comparison of the integral values and standard deviations of MVC in the treatment and experimental groups.

**Figure 10 sensors-24-04673-f010:**
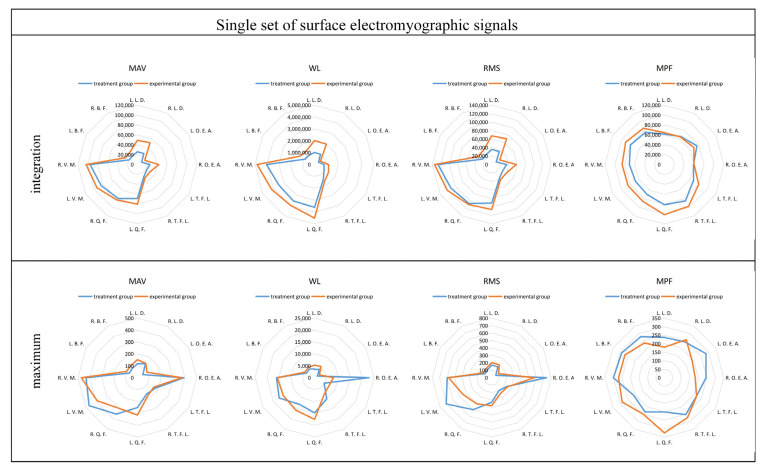
Comparison of four eigenvalues of the same tester.

**Figure 11 sensors-24-04673-f011:**
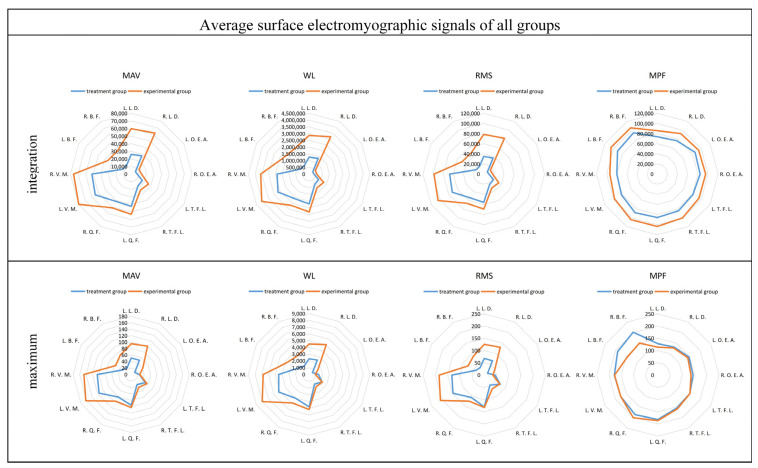
Comparison of the four eigenvalues after averaging across all testers.

**Table 1 sensors-24-04673-t001:** Name of the model selected muscles.

Serial Number	Muscle Name	Position in the Human Body	Remarks
1	biceps femoris (B. F.)	posterolateral thigh	lower limb muscle
2	vastus medialis (V. M.)	anteromedial thigh	
3	vastus lateralis (V. L.)	anterolateral thigh	
4	quadriceps femoris (Q. F.)	anterior middle thigh	
5	tensor fasciae latae (T. F. L.)	upper thigh anterolateral	
6	latissimus dorsi (L. D,)	back and chest posterolateral	waist and abdominal muscles
7	obliquus externus abdominis (O. E. A.)	lateral abdomen	
8	trapezius (T.)	neck and back	upper limb muscles
9	deltoid (D.)	upper shoulder side	
10	triceps brachii (T. B.)	posterior upper arm	
11	biceps brachii (B. B.)	anterior upper arm	

**Table 2 sensors-24-04673-t002:** Correspondence between muscles and marker points.

sEMG Sensor Number	Muscle Name	Position in the Human Body	Position of Electrodes
1	Left latissimus dorsi (L. L. D.)	back and chest posterolateral	On the left side, the center line from the lower chest to the waist, to the humerus behind the armpit.
2	Right latissimus dorsi (R. L. D.)	back and chest posterolateral	On the right side, the center line from the lower chest to the waist, to the humerus behind the armpit.
3	Left obliquus externus abdominis (L. O. E. A.)	lateral abdomen	The left side starts from the outside of the lower eight ribs and ends at the front of the iliac crest.
4	Right obliquus externus abdominis (R. O. E. A.)	lateral abdomen	The right side starts from the outside of the lower eight ribs and ends at the front of the iliac crest.
5	Left tensor fasciae latae (L. T. F. L.)	upper thigh anterolateral	Left anterior superior iliac spine, muscular belly wrapped between two layers of fascia lata.
6	Righ tensor fasciae latae (R. T. F. L.)	upper thigh anterolateral	Right anterior superior iliac spine, muscular belly wrapped between two layers of fascia lata.
7	Left quadriceps femoris (L. Q. F.)	anterior middle thigh	At 50% on the line from the anterior spina iliaca superior to the superior part of the patella on the left leg.
8	Right quadriceps femoris (R. Q. F.)	anterior middle thigh	At 50% on the line from the anterior spina iliaca superior to the superior part of the patella on the right leg.
9	Left vastus medialis (L. V. M.)	anterolateral thigh	At 80% on the line between the anterior spina iliaca superior and the joint space in front of the anterior border of the medial ligament on the left leg.
10	Right vastus medialis (R. V. M.)	anterolateral thigh	At 80% on the line between the anterior spina iliaca superior and the joint space in front of the anterior border of the medial ligament on the right leg.
11	Left biceps femoris (L. B. F.)	posterolateral thigh	At 50% of the line between the left leg ischial tuberosity and the lateral epicondyle of the tibia.
12	Right biceps femoris (R. B. F.)	posterolateral thigh	At 50% of the line between the right leg ischial tuberosity and the lateral epicondyle of the tibia.

**Table 3 sensors-24-04673-t003:** Noise sources and concentrated frequency distribution.

Main Sources of Interference Noise	Frequency Range	Main Drain Effects
power-line interference	50 Hz	The signal-to-noise ratio of EMG signal is reduced.
Inherent noise of equipment	0~30 Hz	Electromyographic signal produces zero drift.
low frequency noise	0~30 Hz	It is one of the main noise interference components.
Human body noise	3~17 Hz	It is one of the main noise interference components.
baseline drift	0~20 Hz	The interference generated during the acquisition process

## Data Availability

The datasets used or analyzed during the current study are available from the corresponding author on reasonable request.

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
