# Peer review of "Positional Analysis of Assisting Muscles for Handling-Assisted Exoskeletons"

_sensors, 2024, doi:10.3390/s24144673_

Round 1
Reviewer 1 Report
Comments and Suggestions for Authors
The article entitled "Positional analysis of assisting muscles for handling-assisted exoskeleton" presents an investigation into the biomechanical characteristics of the human muscle system. The authors analyse the biomechanics of human movements to better design the handling-assisted exoskeleton. They employ the Anybody Modeling System (AMS) and surface electromyography (sEMG) experiments to identify the muscles that need support during handling.
However, it is unfortunate that this study has a very low level of overall merit, particularly in terms of presentation quality, scientific rigor, and significance of content. I regret to inform you that I must reject the article due to serious flaws, the need for additional experiments, and incorrect research methodologies. Notably, the authors do not include patient data, which would provide more realistic insights than constructing models simulating.
The manuscript should be significantly revised before re-submission. The following issues must be addressed:
• The entire text of the manuscript should be thoroughly checked and improved from an editorial perspective. The authors consistently omit spaces in sentences before citing the literature (before brackets), among other issues. The text contains a reference to Zajac (l. 257) and the list lacks a source of literature!
• The authors should more clearly present the main objectives of the study, ensuring they align with the article's content. The current focus is solely on the biomechanics of lower limb movements during standing and squatting (with and without weight) in the context of exoskeleton design.
• EMG analysis only covers the lower limb and torso muscle groups, but Table 1 includes upper limb muscles within the model. This discrepancy is not well justified in the context of the study.
• The study involves two groups: experimental and control. However, it should be clarified why these groups consist of the same individuals rather than different subjects. Furthermore, the study only includes healthy young men, limiting the generalizability of the results to other demographic groups.
• The methodology is poorly described. The section "Materials and Methods" includes guidelines (lines 132–146) but lacks detailed descriptions of AMS modeling and sEMG experiments. The study should provide a comprehensive description of the research methodology, including the apparatus used, boundary conditions, and input data for AMS.
• It is unclear whether the exercises analyzed include standing up, squatting, or both. How was the range of motion (ROM) measured? Were the moments different for standing up and squatting? If analyzed separately, the results should be presented accordingly.
• It is crucial to verify whether the AMS simulation accurately reflects the experimental studies. For example, Figure 1 shows a model in an unnatural squat position. The muscle numbers in Table 1 do not match the muscle list and sensor numbers in Table 2.
• Formulas (1) and (2) are missing.
• Figure 6, showing sEMG signals from 12 muscles, uses muscle numbers instead of names as in previous Table 3 and Figure 4.
• For Figure 7, the authors should include the raw EMG signal for the same muscle and exercise phase.
• The comparative analysis of results between groups is inadequate and needs improvement. There was only one group that performed two tests (with and without weight). The AMS results compared with sEMG should only cover the examined muscle groups of the lower limb and torso.
• The authors should also discuss the validation of their findings, especially concerning AMS.
• The conclusions and results extend beyond the scope of the study. The authors should explain and relate their findings to the existing literature.
• The authors could provide a more detailed discussion of the study's limitations and the impact of varying experimental conditions. They should also propose and justify future research directions.
The authors should address and clarify the above issues in the revised manuscript. They need to restate the discussion, incorporating these comments and addressing limitations. Additional explanations and expansions could enhance the clarity and transparency of the study.
Recommendation: I reject this article for publication in its present form.
Author Response
Comment 1:The entire text of the manuscript should be thoroughly checked and improved from an editorial perspective. The authors consistently omit spaces in sentences before citing the literature (before brackets), among other issues. The text contains a reference to Zajac (l. 257) and the list lacks a source of literature!
Response 1: Thank you for pointing this out. I agree with this comment. Changes were made in the text to address the formatting issues raised. Regarding the issue of "Zajac", he is not a literature citation, but rather a reference to his theory within the AMS software, and is not directly related to the use of the citation in this study.
Comment 2: The authors should more clearly present the main objectives of the study, ensuring they align with the article's content. The current focus is solely on the biomechanics of lower limb movements during standing and squatting (with and without weight) in the context of exoskeleton design.
Response 2:Thank you for pointing this out. I agree with this comment. For the content of the research purpose of this paper, the relevant content is written in the first and last paragraphs of the introduction.
Comment 3:EMG analysis only covers the lower limb and torso muscle groups, but Table 1 includes upper limb muscles within the model. This discrepancy is not well justified in the context of the study.
Response 3:Thank you for pointing this out. I agree with this comment. In response to the question raised about the selection of muscles being different, the question is explained in lines 237-241. sEMG experiments for muscle selection, taking into account the upper limb muscle groups in the case of hand weight-bearing will certainly be the case of larger sur-face EMG signals, do not need the experimental process on which the muscle activation degree is calculated again in the case.
Comment 4:The study involves two groups: experimental and control. However, it should be clarified why these groups consist of the same individuals rather than different subjects. Furthermore, the study only includes healthy young men, limiting the generalizability of the results to other demographic groups.
Response 4: Thank you for pointing this out. I agree with this comment. This experiment was an experiment conducted by 10 healthy men. The experiment consisted of a movement analysis for the handling process. The biomechanics of the human muscles during the lifting process were mainly analysed, because this part of the experiment does not have much relevance to the gender and age of the subjects, and the reliability of the results will not be affected by the fact that the experimenters were only young males.
Comment 5:The methodology is poorly described. The section "Materials and Methods" includes guidelines (lines 132–146) but lacks detailed descriptions of AMS modeling and sEMG experiments. The study should provide a comprehensive description of the research methodology, including the apparatus used, boundary conditions, and input data for AMS.
Response 5:Thank you for pointing this out. I agree with this comment. The description of the relevant elements is in lines 196-199.
Comment 6: It is unclear whether the exercises analyzed include standing up, squatting, or both. How was the range of motion (ROM) measured? Were the moments different for standing up and squatting? If analyzed separately, the results should be presented accordingly.
Response 6: Thank you for pointing this out. I agree with this comment. This paper is a biomechanical analysis of the whole process of the lifting movement and does not differentiate between standing and squatting.
Comment 7: It is crucial to verify whether the AMS simulation accurately reflects the experimental studies. For example, Figure 1 shows a model in an unnatural squat position. The muscle numbers in Table 1 do not match the muscle list and sensor numbers in Table 2.
Response 7: Thank you for pointing this out. I agree with this comment. The muscles in Tables I and II are not exactly the same. In response to the question raised about the selection of muscles being different, the question is explained in lines 237-241. sEMG experiments for muscle selection, taking into account the upper limb muscle groups in the case of hand weight-bearing will certainly be the case of larger sur-face EMG signals, do not need the experimental process on which the muscle activation degree is calculated again in the case.
Comment 8: Formulas (1) and (2) are missing.
Response 8:Thank you for pointing this out. I agree with this comment. Display issues with the article were revised in the manuscript
Comment 9: Figure 6, showing sEMG signals from 12 muscles, uses muscle numbers instead of names as in previous Table 3 and Figure 4.
Response 9: Thank you for pointing this out. I agree with this comment. Figure 6 was modified in the manuscript.
Comment 10: For Figure 7, the authors should include the raw EMG signal for the same muscle and exercise phase.
Response 10: Thank you for pointing this out. I agree with this comment. The raw EMG signals of the same muscles during the movement phase are shown in Figure 5.
Comment 11:The comparative analysis of results between groups is inadequate and needs improvement. There was only one group that performed two tests (with and without weight). The AMS results compared with sEMG should only cover the examined muscle groups of the lower limb and torso.
Response 11:Thank you for pointing this out. I agree with this comment. In response to a related question, in lines 436-443 of the manuscript, the relevant content is depicted.
Comment 12:The authors should also discuss the validation of their findings, especially concerning AMS.
Response 12:Thank you for pointing this out. I agree with this comment. In response to a related question, in lines 472-478 of the manuscript, the relevant content is depicted.
Comment 13:The conclusions and results extend beyond the scope of the study. The authors should explain and relate their findings to the existing literature.
Response 13:Thank you for pointing this out. I agree with this comment. In response to a related question, in lines 444-461 of the manuscript, the relevant content is depicted
Comment 14: The authors could provide a more detailed discussion of the study's limitations and the impact of varying experimental conditions. They should also propose and justify future research directions.
Response 14:Thank you for pointing this out. I agree with this comment. In response to a related question, in lines 475-485 of the manuscript, the relevant content is depicted
Reviewer 2 Report
Comments and Suggestions for Authors
In this study, AnyBody modeling system (AMS) was used to simulate and analyze the motion state of muscle during human transport. Combined with surface electromyography (sEMG) experiment, the specific analysis and verification were carried out to get the muscle position needed to assist the human body in the process of transport. This paper points out that it is reasonable and effective to combine surface EMG with musculoskeletal model simulation to analyze the muscle condition during human movement. In this study, AMS simulation results and sEMG data are combined together to analyze the biomechanics of the human body under specific movements, which makes the analysis results more convincing. Overall, this paper is interesting. Following issues should be well addressed before recommending for publication. 1. Check the spelling principles of the words in Figure 2. 2. What is the basis for the selection of muscles tested in Table 2? 3. Eq 1 and Eq 2 cannot be shown in the manuscript. 4. The spelling of sEMG in Figure 6 and title 3.3 are wrong. 5. Please improve Figure 6 for clarity and impact. 6. In line 334, xi is incorrectly expressed. 7. Symbols with specific meanings should be italicized ( through lines 345). 8. Please enhance the clarity of Figures 9, 10, and 11 to facilitatereaders' understanding. 9. One sentence in the conclusion (lines 483-491) is so long that it is difficult to understand its meaning. Rewrite these sentences to ensure brevity. 10. In the abstract of this study, it is indicated that to optimally design the human transport auxiliary exoskeleton, a biomechanical analysis of human transport is conducted. Nevertheless, the entire text fails to reflect the exoskeleton experimental analysis. If feasible, application cases of the exoskeleton should be incorporated into the research to assist in validating the correctness of the research content. 11. Whether the explanatory content in the Materials and Methods conforms to the content arrangement of the journal. 12. Judging from the title and abstract, AnyBody should be categorized as the key part. Appropriately adding relevant research experiments would be conducive to enhancing the quality of the article. 13. The theoretical portion of the article could be suitably augmented to facilitate deeper comprehension by readers.

Author Response
Comment 1: Check the spelling principles of the words in Figure 2.
Response 1: Thank you for pointing this out. I agree with this comment. I don't see a problem with the description in response to Figure 2, so if any of the words are spelled incorrectly could you please clarify again.
Comment 2:What is the basis for the selection of muscles tested in Table 2?
Response 2: Thank you for pointing this out. I agree with this comment.In response to the question raised about the selection of muscles being different, the question is explained in lines 237-241. sEMG experiments for muscle selection, taking into account the upper limb muscle groups in the case of hand weight-bearing will certainly be the case of larger sEMG signals, do not need the experimental process on which the muscle activation degree is calculated again in the case.
Comment 3: Eq 1 and Eq 2 cannot be shown in the manuscript.
Response 3: Thank you for pointing this out. I agree with this comment. Display issues with the article were revised in the manuscript.
Comment 4: The spelling of sEMG in Figure 6 and title 3.3 are wrong.
Response 4: Thank you for pointing this out. I agree with this comment. Display issues with the article were revised in the manuscript.
Comment 5:Please improve Figure 6 for clarity and impact.
Response 5: Thank you for pointing this out. I agree with this comment. Figure 6 was modified in the manuscript.
Comment 6: In line 334, xi is incorrectly expressed.
Response 6: Thank you for pointing this out. I agree with this comment. Display issues with the article were revised in the manuscript.
Comment 7: Symbols with specific meanings should be italicized ( through lines 345).
Response 7: Thank you for pointing this out. I agree with this comment. Display issues with the article were revised in the manuscript
Comment 8: Please enhance the clarity of Figures 9, 10, and 11 to facilitate readers' understanding.
Response 8: Thank you for pointing this out. I agree with this comment. Display issues with the article were revised in the manuscript
Comment 9:One sentence in the conclusion (lines 483-491) is so long that it is difficult to understand its meaning. Rewrite these sentences to ensure brevity.
Response 9: Thank you for pointing this out. I agree with this comment. In response to a related question, in lines435-460 of the manuscript, the relevant content is depicted.
Comment 10: In the abstract of this study, it is indicated that to optimally design the human transport auxiliary exoskeleton, a biomechanical analysis of human transport is conducted. Nevertheless, the entire text fails to reflect the exoskeleton experimental analysis. If feasible, application cases of the exoskeleton should be incorporated into the research to assist in validating the correctness of the research content.
Response 10: Thank you for pointing this out. I agree with this comment. However, I believe that biomechanical analysis should be a problem that should be solved before the design of the exoskeleton, and subsequent research will be carried out on the exoskeleton, and the current research is only at the stage of biomechanical research before design.
Comment 11:Whether the explanatory content in the Materials and Methods conforms to the content arrangement of the journal.
Response 11: Thank you for pointing this out. I agree with this comment. Formatting and display issues were revised in the manuscript
Comment 12:Judging from the title and abstract, AnyBody should be categorized as the key part. Appropriately adding relevant research experiments would be conducive to enhancing the quality of the article.
Response 12: Thank you for pointing this out. I agree with this comment. In response to a related question, in lines162-174 of the manuscript, the relevant content is depicted.
Comment 13:The theoretical portion of the article could be suitably augmented to facilitate deeper comprehension by readers.
Response 13: Thank you for pointing this out. I agree with this comment. Display issues with the article were revised in the manuscript. Some theoretical descriptions of AMS were added to the manuscript
Reviewer 3 Report
Comments and Suggestions for Authors
The study analyzes 11 muscles that assist in the grasping movement across different postures.
The study is relevant and novel, as it reflects how the control of force in hand movements is dependent on the person's spatial posture.
While it incorporates a classical analysis of sEMG, it would have been interesting to observe the behavior of these muscles not only at the activation level but also over time by analyzing fatigue. Nevertheless, the results are significant.
I would like more details on the MVC procedure for each muscle. How were the sensors calibrated based on their isometric contractions? This is to verify that the procedure was carried out correctly.
Additionally, figures 9, 10, and 11 have very small size numbers. Please modify the format of these images. Remember that this journal has no page limit, so you could place them on a full page without any issues.
Author Response
Comment 1: I would like more details on the MVC procedure for each muscle. How were the sensors calibrated based on their isometric contractions? This is to verify that the procedure was carried out correctly.
Response 1: Thank you for pointing this out. I agree with this comment. On the subject of if you calibrate the position of the sensors, I started by finding the approximate position of each muscle in the body. Then use the real-time values from the sEMG sensors to find the specific location where the experimenter's changes are more pronounced during exercise. I don't know if I have answered your question, if there are still questions, please feel free to continue to communicate.
Comment 2: Additionally, figures 9, 10, and 11 have very small size numbers. Please modify the format of these images. Remember that this journal has no page limit, so you could place them on a full page without any issues.
Response 2: Thank you for pointing this out. I agree with this comment. Corresponding issues of the article were revised in the manuscript.
Round 2
Reviewer 1 Report
Comments and Suggestions for Authors
The revised article still requires corrections. The authors agreed with all of my comments, but not all of them were corrected. First of all, the study involves two groups: experimental and treatment (before: control). Both groups consist of the same individuals rather than different subjects. Therefore, the study cannot be divided into two groups, but this study should be divided into two experiments (two tests): without weight and weighted with 10kg (unweighted and weighted handling).
Moreover, the analyzed handling process (experimental process) should be more precisely described. What kind of movements have it included: standing - squatting - standing up? What was the range/phases of motion measured? Moments of standing up and squatting are different, and different muscles take part. Therefore, the results should be presented according to range/phases of motion and analyzed adequately.
What is the number of steps included in the range/phases of motion during inverse dynamic analysis (1000)? Figure 4 and Figure 8 presented 2000 steps, but the results of sEMG presented in Figures 5 and 6 are in time (s). So, the authors should explain how many steps are contained in one second or how many seconds last one step. It is easy to do if the authors divide the whole movement into ranges/phases of motion.
The authors could extend and provide a more detailed discussion of the study and the authors should address and clarify the above issues in the revised manuscript.
Recommendation: This article needs major revision (improvement of some experiments) and re-reviewing before publication.
Author Response
Comment 1: First of all, the study involves two groups: experimental and treatment (before: control). Both groups consist of the same individuals rather than different subjects. Therefore, the study cannot be divided into two groups, but this study should be divided into two experiments (two tests): without weight and weighted with 10kg (unweighted and weighted handling).
Response 1: Thank you for pointing this out. I disagree with this comment. The control and experimental groups should be set up so that different tests are performed on the same subject, thus creating a comparison. Not that it should be split into two tests as you suggest.
Comment 2:Moreover, the analyzed handling process (experimental process) should be more precisely described. What kind of movements have it included: standing - squatting - standing up? What was the range/phases of motion measured? Moments of standing up and squatting are different, and different muscles take part. Therefore, the results should be presented according to range/phases of motion and analyzed adequately.
Response 2: Thank you for pointing this out. I have added a detailed description of the experimental procedure in lines 186-188 of the manuscript. However, this study is analysing the entire handling process, and in order to guide the design of the exoskeleton, it does not divide the handling process into stages.
Comment 3:What is the number of steps included in the range/phases of motion during inverse dynamic analysis (1000)? Figure 4 and Figure 8 presented 2000 steps, but the results of sEMG presented in Figures 5 and 6 are in time (s). So, the authors should explain how many steps are contained in one second or how many seconds last one step. It is easy to do if the authors divide the whole movement into ranges/phases of motion.
Response 3: Thank you for pointing this out. I added the settings for the steps and the reason why there are 2,000 entries in the data on lines 192-195 of the manuscript. However, this study is looking at the handling manoeuvres as a whole, so there is no stage division.
Comment 4:The authors could extend and provide a more detailed discussion of the study and the authors should address and clarify the above issues in the revised manuscript.
Response 4: Thank you for pointing this out. I have added a discussion of the follow-up research on designing exoskeletons in this study to lines 506-511 of the manuscript.
